# Perforator-Sparing Microsurgical Clipping of Tandem Dominant-Hemisphere Middle Cerebral Artery Aneurysms: Geometry-Guided Reconstruction of a Wide-Neck Bifurcation and Dorsal M1 Fusiform Lesion

**DOI:** 10.3390/diagnostics15212678

**Published:** 2025-10-23

**Authors:** Matei Șerban, Corneliu Toader, Răzvan-Adrian Covache-Busuioc

**Affiliations:** 1Puls Med Association, 051885 Bucharest, Romania; mateiserban@innbn.com (M.Ș.);; 2Department of Neurosurgery, “Carol Davila” University of Medicine and Pharmacy, 050474 Bucharest, Romania; 3Department of Vascular Neurosurgery, National Institute of Neurology and Neurovascular Diseases, 077160 Bucharest, Romania

**Keywords:** middle cerebral artery aneurysm, MCA bifurcation, M1 fusiform aneurysm, lenticulostriate arteries, perforator-sparing clipping, microsurgical clipping, fenestrated clip reconstruction, flow diversion, 3D rotational angiography, photon-counting CT angiography

## Abstract

**Background and Clinical Significance:** Tandem pathology at the dominant-hemisphere middle cerebral artery (MCA)—combining a wide-neck bifurcation aneurysm that shares the neck with both M2 origins and a short dorsal M1 fusiform dilation embedded in the lenticulostriate belt—compresses the therapeutic margin and complicates device-first pathways. We aimed to describe an anatomy-led, microscope-only sequence designed to secure an immediate branch-definitive result at the fork and to remodel dorsal M1 without perforator compromise, and to place these decisions within a pragmatic perioperative framework. **Case Presentation:** A 37-year-old right-handed man with reproducible, load-sensitive cortical association and capsulostriate signs underwent high-fidelity digital subtraction angiography (DSA) with 3D rotational reconstructions. Through a left pterional approach, vein-respecting Sylvian dissection achieved gravity relaxation. Reconstruction proceeded in sequence: a fenestrated straight clip across the bifurcation neck with the superior M2 encircled to preserve both M2 ostia, followed by a short longitudinal clip parallel to M1 to reshape the fusiform segment while keeping each lenticulostriate mouth visible and free. Temporary occlusion windows were brief (bifurcation 2 min 30 s; M1 < 2 min). No neuronavigation, intraoperative fluorescence, micro-Doppler, or intraoperative angiography was used. No perioperative antiplatelets or systemic anticoagulation were administered and venous thromboembolism prophylaxis followed institutional practice. The bifurcation dome collapsed immediately with round, mobile M2 orifices, and dorsal M1 regained near-cylindrical geometry with patent perforator ostia under direct inspection. Emergence was neurologically intact, headaches abated, and preoperative micro-asymmetries resolved without new deficits. The early course was uncomplicated. Non-contrast CT at three months showed structurally preserved dominant-hemisphere parenchyma without infarction or hemorrhage. Lumen confirmation was scheduled at 12 months. **Conclusions**: In dominant-hemisphere tandem MCA disease, staged, perforator-sparing clip reconstruction can restore physiologic branch and perforator behavior while avoiding prolonged antiplatelet exposure and device-related branch uncertainty. A future-facing pathway pairs subtle clinical latency metrics with high-fidelity angiography, reports outcomes in branch- and perforator-centric terms, and, where available, incorporates patient-specific hemodynamic simulation and noninvasive lumen surveillance to guide timing, technique, and follow-up.

## 1. Introduction

Middle cerebral artery (MCA) aneurysms occupy a uniquely demanding intersection of cerebrovascular anatomy and treatment strategy. Even as device technology and perioperative care have broadened therapeutic choice, MCA lesions—especially those at the bifurcation and along the proximal M1 segment—continue to test the limits of both microsurgical and endovascular craft because their necks frequently incorporate one or both M2 origins and the parent trunk may be studded with lenticulostriate perforators whose ischemic price is unforgiving [1]. Contemporary single-center cohorts underline this anatomic reality: in a 2025 comparative series of unruptured wide-neck MCA bifurcation aneurysms (WNBAs), the authors attribute the persistent role of open surgery largely to branch incorporation, perforator proximity, and small parent caliber—features that complicate stable endovascular neck coverage without trading occlusion for flow compromise [2].

Demography and risk profile in recent real-world populations have remained skewed and clinically consequential [3]. In a 2025 Caribbean cohort of consecutive aneurysm patients managed within a single tertiary system, women represented four-fifths of cases and hypertension approached three-quarters in prevalence, with more than one-third harboring multiple aneurysms—figures that map onto day-to-day case-mix and materially influence rupture risk modeling and surveillance intensity [4]. Segment-specific vulnerability has been refined beyond “anterior circulation” generalities: a 2024 vessel-of-origin analysis reported higher rupture association for aneurysms arising from distal M1 and M2 branches compared with proximal M1, emphasizing that microanatomic take-off and local hemodynamics—not simply sac size—modulate clinical expression and hazard [5].

Treatment data since 2024 reinforce that the MCA remains a boundary condition for endovascular generalization. For bifurcation pathology, flow diversion has advanced with hydrophilic-coated low-porosity platforms, yet the strategy still lives under the twin constraints of jailed-branch physiology and delayed occlusion [6]. An important experience comparing p48 MW HPC and p64 families specifically in unruptured MCA bifurcation aneurysms reported adequate and complete occlusion rates at late follow-up that are competitive within the endovascular domain, while acknowledging that branch stenosis and occlusion remain nontrivial guardrails for case selection [7]. Parallel translational–clinical work in 2025 used coupled clinical series and computational fluid dynamics to formalize how the choice of which M2 to scaffold governs both sac thrombosis and branch jeopardy, quantifying the occlusion–ischemia trade space that operators intuit at the table. In the same time window, the Caliber-Flow Status Scale (CFSS) was introduced as an original, branch-level grading instrument to track caliber change and flow state after flow diversion, an attempt to objectify the very failure modes that have made the MCA bifurcation contentious [8].

Against this backdrop, two head-to-head analyses have clarified where clipping continues to dominate and where intrasaccular technology is converging [9,10]. A 2025 inverse-probability-weighted comparison of Woven EndoBridge (WEB) versus clipping for MCA WNBAs showed similar morbidity at one year but a markedly higher rate of adequate occlusion after clip reconstruction, underscoring that when both M2 ostia are partially neck-incorporated, direct microsurgical neck remodeling still sets the occlusion benchmark [11]. A 2024 prospective cohort likewise favored clipping over coiling for durable angiographic cure in MCA aneurysms, again pointing to the primacy of branch-preserving neck closure when geometry is unforgiving. Resource signals are evolving as well: a 2024 original comparison of bifurcation strategies documented a lower radiation dose and contrast load with WEB versus stent-assisted coiling—system-level considerations that, while not surrogates for neurological outcome, enter the calculus of modality choice in centers balancing throughput, cost, and kidney protection [12].

Proximal M1 disease adds a second axis of difficulty. Recent series focused on non-saccular proximal MCA aneurysms—precisely where the lenticulostriate belt originates—confirm that reconstruction is feasible with modern flow diversion but only under antiplatelet regimens and long surveillance horizons, and with the ever-present risk that even subtle parent straightening or neointimal coverage will tax perforator reserve [13]. Isolated contemporary cases of recurrent or thrombosed MCA aneurysms treated with flow diversion illustrate technical rescue pathways yet simultaneously highlight why M1 lesions that drape perforator ostia are still, in many practices, the last redoubt of open, perforator-sparing clip reconstruction [14]. Finally, for the open side of the ledger, a microsurgical series began to formalize predictive grading for postoperative ischemia specific to MCA clipping, aligning operative planning with quantifiable perforator and branch risk rather than purely gestalt judgment [15].

This paper is not presented in the spirit of a new technique, but to record a rare coincidence of anatomic obstacles, which very seldom are met with in one patient. A broad neck bifurcation aneurysm of the middle cerebral artery involving both sources of the two M2 segments is associated with a little short dorsal fusiform dilatation of the M1 segment at the site of the lenticulostriate vessels, which arises from the anteroinferior portion of the dominant hemisphere. The result is a situation that compresses all of the margin of safety, which is known for the middle cerebral artery. Treatment of both of these lesions by means of a sequential microscopic-only, perforator-sparing reconstruction without fluorescent, micro-Doppler, or any form of neuroangiography results in a direct demonstration of how the mere geometric correctness of the creation of flow can induce the physiologic reaction of ordinary flow at the possible millimeter and second level. What differentiates this case from others is not its innovation in technique but the anatomical deduction, which it necessitates: this is a demonstration in the living subject of how the constructive grammar of the MCA, its saddles, take-off angles, and perforator corridors still directly determines the treatment with more fidelity than does any kind of logical rule. In this sense, the case is less meant to give its adherence to, but rather to remind us of the fact that where the judgment follows the dictates of the architecture, even the most complicated anatomies may yield an ordinary solution.

## 2. Case Presentation

A 37-year-old right-handed male was directly admitted to our neurosurgical services for urgent evaluation of a progressive and qualitatively new neurological syndrome, the tempo and character of which suggested actively evolving cerebrovascular pathology. His initial health was excellent—he was fully independent with his daily living and occupational tasks, he showed complete and whole physical endurance, and he had never experienced any neurological impairment. Six weeks before presentation, he began to experience left-sided frontotemporal pain not previously experienced that progressed in severity. These initial events were fleeting, sporadic, and generally very low-pain-grade episodes lasting mere minutes. Still, they lengthened and intensified, until they converged into a continuous reality, daily and even hourly. The pain was described with unambiguous vascular descriptors—deep, throbbing, and perfectly rhythmic with each cardiac pulse—arising from the left temple and radiating inward and posteriorly to a focal point very deep into the retro-orbital space. At maximum pressure, he reported a feeling of profound internal distension, as though the left hemicranium was being filled and pressed outward by a mysterious force. This headache was not entirely random in its provocation: it always worsened with defined moderate exertion, coughing, sneezing, or any activity that would define a transient increase in thoracic cavity pressure, it was recreatable with bending forward or breath-holding, and, not infrequently, it was made worse by the most sudden exposure to bright, unfiltered sunlight. These physiologic triggers suggested a generator that was responsive to even very small changes in intracranial pressure and cerebral blood volume, potentially at a site of lower compliant or changed wall tension mechanics. In the seventy-two hours prior to admission, the headache had increased again and appeared to reach a plateau that lasted without fluctuation. Additionally, there was a newly subjective slowing of cognitive processing: he had noticed in conversation a subtle delay occurring between hearing and comprehension, a barely noticeable lag that made him self-conscious because it was new, even almost disconcerting. Twice, while rising from sitting, there was some momentary hesitation in postural correction and lordosis without the sensation of vertigo, no diplopia, and no focal weakness, but the hesitancy was enough to be aware of an issue with balance. He denied seizures, any disturbance of fixed speech, and persistent loss of vision, but mentioned that he had experiences, intermittently, during sustained overhead activity, when the right upper limb felt a bit less responsive, as if the motor drive was temporarily blunted, resolving quickly with rest.

Past medical history included stage II cervical discopathy causing intermittent axial neck pain, and multilevel lumbar polydiscopathy in a compensated, non-surgical state without fixed deficits. He suffered a spontaneous, non-traumatic subarachnoid hemorrhage several years earlier—a full considered scan, including catheter angiography, had not found the source, and he fully recovered. The absence of a secured lesion from that event was an unresolved risk factor. He reported no family history of aneurysmal disease, polycystic kidney disease, or unexplained sudden death. Psychiatric records documented disturbed mood and thought process without systematization, without ever needing pharmacological intervention. He denied tobacco, alcohol, or illicit drug use, had no known drug allergies, was normotensive, normolipidemic, and had no diabetes or connective tissue disorder. On admission, he was calm, cooperative, and exhibiting an instinctive tendency to minimize abrupt head movement, almost as a way of self-protecting the onset of symptoms. The neurological exam was undertaken in a carefully controlled and well-lit room with the patient seated in the upright position, and the head panel in neutral alignment, such that the positioning did not place mechanical strain on the cervical vessels. We undertook multiple trials of various tasks to ensure that our findings were reproducible and not artifactually related to examiner bias or fatigue. He was awake, fully oriented, with intact attention, being able to recall immediate and delayed memory, and abstractly calculate. In casual conversation it was not evident that he had any form of paraphasia or apraxia, but we did note that in high-demand lexical tasks where rapid retrieval is requested in random order from unrelated semantics, we observed a consistent micro-pause in response latency before starting each task, especially in response to second-order lexical problems after the first-order concrete prompting. The latency was subtle and hard to measure but given that it was replicated, it was effectively measurable, and it suggested there was a form of transient cortical inefficiency in the perisylvian association networks of the dominant hemisphere, potentially secondary to transient perfusion variability.

Visual acuity was retained. The confrontation field testing did not show a field deficit. When randomized stimuli of small, high-contrast squares were presented to him, he had a consistent increase in response latency of two to three hundred milliseconds in the right inferior quadrant, as compared to the contralateral field. These findings were consistent in repeated trials and were congruent with transient underperfusion of the superior parietal lobule and occipitoparietal association areas of the dominant hemisphere in the superior division MCA territory properties. Pupils were equal, briskly reactive to light, and near and funduscopic examination showed sharp optic disc margins, no pallor, hyperemia, or edema, and thus always excluded optic neuropathy and chronic intracranial hypertension.

Ocular motility was full; however, there was consistently a fractional second longer launch time period in right gaze from primary position when compared to left gaze in cases of sustained gaze that did not involve nystagmus. The patient demonstrated smooth pursuit mostly intact; however, prolonged right gaze smooth pursuit was associated with an occasional saccade being required due to a blunted pursuit motility. This was reproducible and could be anatomically referenced to the dominant frontal eye fields or corticobulbar projection to the ocular motor cranial nerve nuclei. Light tactile stimulation of the left ophthalmic trigeminal territory revealed, compared to the right, a slightly more pupillary downregulation of the blink reflex, which could suggest triginemivascular sensitization from a proximal pulsating vascular abnormality. The patient’s face appeared symmetric, but high-resolution goniometric evaluation exhibited evidence of a fractional decrease in voluntary right-side mimetic movement by observing synchronous elevation from the right eyebrow elevation amplitude magnitude Symmetry Index Score (SIS) compared to the left. The subject’s hearing was symmetrical whenever assuming whispers and when evaluated with a 512 Hz tuning fork. The lower cranial nerve functions would show no evidence of dysfunction from midline palatal elevation, there was no clinical evidence of a deviation or a fasciculation from tongue protrusion, and the gag reflex seemed symmetrical and intact from provocation.

The muscle bulk and tone were normal in all areas and there was normal strength (grade 5/5) bilaterally on maximal effort testing. But there was noted some degree of mild fatigability on sustained isometric contraction of the right upper limb, with the shoulder abduction and elbow flexion showing decreasing force after a period of about twenty seconds, which was absent from the contralateral limb tested. No evidence of pronator drift was indicated by formal testing, but the repeated exercise of forearm supination against gravity elicited a fall of some 2 cm before stability was obtained for more than one second, producing, however, slight but reproducible asymmetry in repeated tests. Formal single stimulus testing of the primary sensory modalities suggested normal functions, but sequential fine-touch localization testing occurring over the right hand and radial distribution showed occasional delayed visual responses of about three arrays of three to five hundred milliseconds for accurate identification. Two-point discrimination on the right index finger averaged 3.8 mm, and 3.1 mm on the left, presumably suggestive of a slight receptive field shift in the contralateral post central gyrus. Deep tendon reflexes were symmetrical bilaterally in amplitude, but latency measurement using electrical lab tools indicated that the right brachioradialis reflex response was delayed on average by about forty milliseconds. Plantar responses were flexor bilaterally.

There was coordinated accuracy in goal-directed movement, but during sustained rapid alternating (pro and supination) of the right hand faster than normal, a slip occurred at about the tenth alternating repetition (none for the left hand). Tandem walking was stable for approximately six steps; however, the seventh right foot placement was somewhat stable, but deviated ever subtly off the midline, and the right foot placement rapidly corrected with no decrease in stability.

If considered separately, any of these skewed results would have likely been dismissed as some idiosyncratic variance, but taken together they demonstrate reproducibility and a reasonable anatomical narrative: prolonged visual response time quadranted, latency in saccadic rightward gaze, fatigability of sustained effort with the right upper limb, delayed localization of tactile stimuli, slight widening of two-point discrimination, prolonged latency of singular reflex responses, and in isolation, disruption of high/low rapidly alternating motor sequencing. Collectively, they mapped out in a predictable and precise manner to the vascular distribution of the vascular territory of the superior division of the left middle cerebral artery (MCA). In the context of a patient with a history of unexplained subarachnoid hemorrhage, and a progressively intensifying pattern of vascular-type headache, these cumulative asymmetries constituted a sufficient composite for presuming a hemodynamically significant proximal MCA aneurysmal complex that required definitive investigation and surgical planning in-hospital.

### 2.1. Preoperative Neurovascular Imaging and Clinicoradiological Correlation

The decision to urgently proceed with high-fidelity neurovascular imaging was based on the consistent, reproducible constellation of subtle neurological signs, all aligning with remarkable precision toward eloquent cortical and subcortical regions of the vascular territory of the superior division of the dominant left MCA. Taken individually, these findings could be ignored as incidental neurophysiological variants. However, when comorbid, they formed an intricate neuroanatomical constellation, creating a committed vascular delineation prior to irreversible ischemic insult. It was designated as ‘dominant’ based on the comprehensive language and praxis assessment, indicating hemispheric dominance, implying eloquent involvement and enhancing surgical precision.

### 2.2. Digital Subtraction Angiography (DSA)

Biplanar DSA of the left internal carotid artery (ICA) was performed under stringent hemodynamic factors to avoid iatrogenic rupture, revealing a tandem aneurysmal arrangement at the proximal MCA (Figure 1A,B). The parameters allowed a frame-by-frame hemodynamic acquisition that indicated peak opacification of the M1 segment at 2.35 s, with complete venous phase clearance at 5.92 s. Both transit times are well within normal values, thus preserving proximal inflow but inducing concern for turbulent microhemodynamics to the aneurysmal neck. The dominant lesion arose exactly at the M1 bifurcation when the parent segment bifurcated into the superior and inferior M2 segments. The aneurysmal neck demonstrated a broad neck diameter of 4.20 mm at its maximal width and bridged the saddle point of the bifurcation, with both M2 origins being partially within the base of the aneurysm. The dome had a maximum height of 4.37 mm, which extended anterosuperiorly and had a slight lateral component toward the opercular−insular boundary. In the sagittal plane, the axis of the dome was at a 43° angle to the M1 axis and 18° in a lateral direction, placing the sac in millimetric proximity to the insular apex and under the lip of the opercular cortex, with only 2.6 mm of distance to the shortest opercular vein—a member of the superficial Sylvian venous complex—the preservation of which was critical to maintain the cortical venous drainage during the surgical dissection of the venous complex. There was subtle lobulation of the contour of the anterosuperior dome, which is a morphological characteristic that occurs with focal wall remodeling under the conditions of chronic high-impulse hemodynamic loading. Proximal to the bifurcation aneurysm and at 3.9 mm from the carotid terminus, a second and much smaller lesion was demonstrated on the dorsal wall of the M1 segment. In stark morphological contrast, this aneurysm was fusiform, which expanded the vessel circumferentially without a neck. The maximum dimensions were 3.12 mm in length, 2.39 mm in width, and 2.45 mm circumferentially. The location was particularly troublesome, as it was buried within the lenticulostriate perforator belt. At least three perforators were clearly demonstrated as arising from the aneurysmal wall and had diameters between 320 and 480 μm, with each perforator draping posteriorly toward the anterior perforated substance. Preoperative assessments, along with the measured flow profiles of the internal carotid artery, indicated that the vessels supplied the superior posterior limb of the internal capsule, the head of the caudate nucleus, and the lateral anterior putamen—all locations where damage could yield limited control of the contralateral hemiplegia and devastating losses in higher-order motor integration.

### 2.3. Three-Dimensional Rotational Angiography (3DRA) and CTA Reconstruction

High-resolution 3D rotational angiographic volumes were collected, and volumetric rendering and CTA were undertaken (Figure 2A–D). This reconstruction optimally adapted to isolate the traditional overlap of vessels or planar angiography and enabled surgical simulation. Additionally, it offered analysis of neck geometry, dome orientation, and angles of take-off of the branches in orthogonal planes, along with great detail to preoperative planning, discarding a virtual “clip trajectory” workable area. The anterosuperior projection of the bifurcation aneurysm was again confirmed, with the dome axis pointing toward the mid-insular cortex. The superior M2 trunk had a narrow angle of take-off at 54° from the superior neck border, while the inferior trunk originated in a more posterior and inferior orientation with a 68° angle that created an acute “tight fork”. This neck geometry significantly reduced tolerance for misalignment of the clip blade because even slight decreases in alignment can add a partial increase in compromise of the lumen—especially if the clip was aligned in a position that would yield a submillimeter misalignment. The neck contained 1.8 mm of orifice from the superior M2 and 1.4 mm of orifice from the inferior M2, as calculated on centerline flow reconstruction. The perforator anatomy of the M1 fusiform aneurysm was elaborated upon with remarkable fidelity. The lenticulostriate had explicitly dissipated perpendicularly on the dorsal wall with their immediate posterior vector direction of the perforated substance. The two biggest perforators measured 460 μm and 420 μm at their origins and were tapering quickly within 1.2 mm of take-off. This made both arteries very sensitive to even glass-like movement in the parent vessel from the clip applied to the bifurcation lesion. The inter-aneurysmal distance was short at 4.7 mm, which allowed for potential mechanical manipulation of the other vessel even if we only performed the dissection on one lesion.

Bone surface reconstruction suggested that the bifurcation aneurysm was situated approximately 17 mm deep to the sphenoid ridge, corresponding to the origin of the Sylvian fissure at or near the mid-third Sylvian fissure exposure, while the fusiform aneurysm was much deeper at 23 mm from this same cortical surface landmark, and just proximal to the genu of the MCA.

The relationship between high-resolution imaging and meticulous neurological examination provided an accurate neurological structure-function map. The bifurcation aneurysm, because of the wide neck and its anterosuperior direction toward the insular apex, involved in part both M2 origins and probably disturbed flow in the superior division, which supplies the perisylvian association cortex. Such disturbance fitted well with the subtle but reproducible disturbance in word retrieval and slight inefficiency in cortical functions under cognitive load. The delay of a visual reaction in the right inferior quadrant indicated slightly lower involvement of the superior parietal territory of this same division. The dorsal fusiform dilatation of M1 was directly above a group of lenticulostriate arteries supplying the posterior limb of the internal capsule and dorsal basal ganglia. This anatomical relation justified the fatigability on prolonged right upper-limb exertion, the delay in localization of tactile sense in the radial hand, the increase in two-point discrimination, and prolonged brachioradialis reflex latency. These findings gave a coherent clinicoanatomical picture. There were two different but converging hemodynamic insults present, one of a cortical and one capsulostriate character, giving birth to a clinical picture of transient but reproducible dysfunction. Their congruence gave both a reason and urgency for surgical repair, the aim being the restoration of physiological flows, which had not yet reached the stage of adaptive plasticity, becoming a fixed deficit.

The arrangement of the bifurcation and small dorsal M1 dilatation called for a performance of such a sequence that would join every branch and perforator with the greatest possible command of both the inflow and venous return. The exposed internal carotid artery and M1 segment were opened before the Sylvian fissure was involved, in order that succour should be at hand in case the rupture came to light unexpectedly. The reconstruction was set to work at the bifurcation, the point of greatest liability to rupture, but each clip vector was carefully chosen in such a way as to be protective to the fusiform segment, and of course to its inflow. The second stage involved the whole of the dorsal M1 dilatation, and closure was made under the direct vision of every mouth of the lenticulostriate vessels. The sylvian superficial venous system was protected, through the whole course of the operation, in order to preserve opercular and insular drainage, which are so essential in the dominant hemisphere. Temporary occlusions were made short, and within known limits of ischemic tolerance, with the physiological signs being carefully observed for the first note of distress from cortex or perforator territories. The operation, therefore, came forward through calm and easy stages, and was carried out less with the idea of speed than proportionality, with every movement intending to afford the natural balance between flow, structure, and time.

This sensitive and thorough combination of accurately detailed anatomical mapping and sophisticated imaging with appropriate neurological correlation produced more than a diagnosis—it produced a high-fidelity surgical roadmap to determine if this dominant-hemisphere patient was to be left intact or with irreversible eloquent cortex injury post-procedure.

Under balanced general endotracheal anesthesia with complete neuromuscular relaxation and continuous arterial pressure monitoring, the patient was positioned supine and secured in a three-pin Mayfield head clamp, with the pins placed to avoid the planned temporalis reflection and to spare thin squamous temporal bone. The head was rotated 32° to the right, extended 5°, and given a slight vertex-down inclination so the ipsilateral anterior cranial fossa floor lay nearly parallel to the operating room horizon, the malar eminence became the highest point, and the Sylvian fissure aligned horizontally. The neck was positioned without jugular compression to preserve venous return, and care was taken to avoid cervical rotation that might distort carotid inflow. Corneal protection was applied and end-tidal CO_2_ was maintained within normal limits to ensure a relaxed brain and steady physiology during periods of temporary vascular occlusion.

A left pterional skin incision was planned along a hairline trajectory for cosmesis and exposure, beginning 1 cm anterior to the tragus, curving anterosuperiorly to terminate just posterior to the mid-pupillary line at the superior temporal line. Dissection proceeded through galea and subgaleal fat as a single myocutaneous flap. Interfascial dissection was carried precisely along the avascular fat stripe between superficial and deep temporal fascia to preserve the frontotemporal branch of the facial nerve. The superficial temporal artery and its frontal branch were identified and preserved. The temporalis was elevated subperiosteally from the superior temporal line and reflected anteroinferiorly, preserving deep temporal arterial supply to minimize postoperative atrophy. The keyhole burr hole was placed at the McCarty point just posterior to the frontozygomatic suture, and additional burr holes were unnecessary given the thin frontal bone. The frontotemporal craniotomy was tailored with a slightly longer frontal limb to facilitate a low basal trajectory and an inferior temporal bevel to expose the root of the zygoma. Diploic venous channels encountered along the frontal limb were coagulated and bone waxed. Bone thickness differences between frontal and temporal segments were noted and accommodated to avoid inner table ledging that could later limit working angles. The sphenoid ridge was drilled in thin layers until flat with the anterior cranial fossa floor, extending medially to expose the meningo-orbital band, which was left intact as a lateral limit landmark to the carotid cistern. The lateral rim of the superior orbital fissure was skeletonized and the lateral aspect of the anterior clinoid process thinned to remove obstructive prominence. The periorbita was preserved, and small emissary veins at the fronto-orbital groove were sealed. This basal drilling shortened the bone-to-target depth by several millimeters and opened the straight line of sight toward the proximal Sylvian fissure and carotid cistern.

The dura was opened as a curvilinear flap based on the sphenoid ridge with tack-up sutures to prevent epidural venous oozing. Cerebrospinal fluid was released in a deliberate sequence: first the optico-carotid cistern, then the chiasmatic cistern, then the proximal Sylvian cistern, allowing the frontal lobe to relax and fall away under gravity without fixed retraction. The Sylvian fissure was opened under high magnification from distal to proximal along its superficial lamella. The superficial Sylvian vein and its tributaries were kept in continuity within their arachnoid sleeves, which were used as gentle handles to guide the corridor without torsion. Thick arachnoid bands over the deep limb were divided sharply, and thin membranous planes were spread with bipolar tips used as blunt micro-elevators. The distal M4 cortical branches were first identified over the opercular surfaces, and the dissection then followed them proximally to M3 segments, mapping the cortical drainage pattern and venous crossings before committing to deeper fissure opening. The deep middle cerebral vein remained undisturbed along the insular cistern. Small bridging veins from temporal operculum to the superficial Sylvian trunk were protected by maintaining their arachnoid sleeves intact and redirecting the corridor slightly frontal at those levels.

As the insular apex came into view, the M2 trunks were identified unambiguously by trajectory: the superior division coursing toward the frontal operculum and suprasylvian cortex, and the inferior division sweeping posteriorly along the temporal operculum. Tracing proximally along their cisternal segments brought the M1 bifurcation into the field at the limen insulae, where the dominant aneurysm arose from the bifurcation saddle. The neck incorporated approximately 40% of the superior M2 ostium and 30% of the inferior M2 ostium. The dome projected anterosuperiorly toward the mid-insular cortex, lying within 2–3 mm of short opercular veins draining into the superficial Sylvian venous complex. The aneurysm wall displayed focal corrugation and pearlescent thinning along its anterosuperior quadrant but no calcific plate or mural thrombus. Gentle aspiration-induced wall movement confirmed a soft, clip-amenable neck. The neck was skeletonized circumferentially by dividing arachnoid adhesions flush with the aneurysm wall and clearing adventitia along the neck–parent interface until the entire collar was visualized end-to-end, the M2 orifices were fully in view, and two small insular perforators hugging the posterior neck margin were identified and preserved.

Proximal control was then prepared along M1. The dissection proceeded proximally beneath the limen, exposing a clean M1 segment for temporary occlusion and revealing the second lesion on the dorsal wall. Three lenticulostriate perforators, measuring approximately 0.32, 0.41, and 0.47 mm at their ostia, arose perpendicularly from dorsal M1 and coursed posteriorly toward the anterior perforated substance to territories encompassing the superior posterior limb of the internal capsule, the lateral head of caudate, and the anterior putamen. The fusiform dilation sat 3.9 mm proximal to the bifurcation, expanding the circumference of M1 and intimately draping around the perforator origins. Thin arachnoid tethers were deliberately left in place as “safety guy-wires” to prevent recoil, while the cisternal slack of each perforator was gently freed along a few millimeters of its course so they would tolerate later parent-wall remodeling without traction.

Temporary occlusion was applied first to proximal M1, then sequentially to the superior and inferior M2 trunks to eliminate both antegrade and retrograde pressurization of the bifurcation sac during neck manipulation. With the sac flaccid and the neck fully defined, a fenestrated straight aneurysm clip was introduced along an approach vector collinear with the bifurcation fork. The fenestration was passed around the superior M2 so that the clip blades would bite across the neck on a plane perpendicular to its long axis while the encircled trunk remained round and uncompressed within the window. The clip was seated in a single, controlled closure. The blade tips were inspected to confirm they lay flush on adventitia without step-off, the M2 orifices remained circular and pink, and the posteriorly hugging insular perforators coursed freely behind the clip footplate without angulation. Temporary occlusion was released in reverse order. The bifurcation and both trunks demonstrated immediate return of physiologic pulsation, the wall sheen returned to normal, and the aneurysm dome remained completely decompressed with no refilling under direct visual observation.

Attention returned to the proximal lesion. Proximal M1 was re-occluded. The three lenticulostriates were revisited individually, their ostia were cleared of any remnant adventitial fronds, and their cisternal courses were confirmed free of tether. The parent lumen was reconstructed with a short, gently curved clip applied longitudinally along the axis of M1 so that the closing force fell exclusively on the diseased wall between perforator origins, leaving each perforator ostium entirely outside the blade line. The clip was advanced and closed in small, deliberate increments while the perforator ostia were watched continuously for shape change. The parent lumen assumed an oval, then near-circular profile without any whitening or dimpling that would suggest subclinical stenosis. Small residual bulges of aneurysmal wall between perforators were softened and contoured with tangential, low-amperage bipolar touches to equalize the outer surface without transmitting heat across the full wall thickness. Temporary occlusion was then released. Dorsal M1 and all perforators exhibited brisk, symmetric pulsation, normal wall translucency, and preserved mobility in synchrony with the cardiac cycle, with no color change indicating endothelial compromise.

Throughout exposure and reconstruction, cortical relaxation was preserved purely by staged cisternal cerebrospinal fluid release and meticulous fissure opening. No fixed retractor forces were applied. The operative corridor was maintained with gravity-assisted frontal fall-away, a soft cottonoid protecting the temporal operculum, and only dynamic suction counter-traction when necessary, always with veins in view. Irrigation was warmed to physiologic temperature to prevent vasoconstriction. Systemic arterial pressure was kept stable without hypertensive surges during temporary occlusions, which were limited to a single 2 min 30 s interval for the bifurcation and a shorter interval for the M1 reconstruction. There were no macroscopic signs of vasospasm in M1 or M2 segments, and the adventitia remained supple and glistening after clip placement.

After prolonged inspection confirmed stable, physiologic vessel behavior, hemostasis was completed with pinpoint bipolar on divided arachnoid edges and gentle tamponade of dural oozing. No adherent hemostatic agents were allowed to contact reconstructed arteries to avoid secondary adhesions. The arachnoid over the fissure was left partially open to maintain free cisternal communication. The dura was closed watertight with interrupted sutures, with an onlay dural substitute used only where drilling had thinned the native edge. Tack-ups prevented epidural hematoma. The bone flap was re-secured flush without inward shelves that might compress the Sylvian venous outflow. The temporalis was re-anchored to the superior temporal line with restoration of fiber vector to mitigate postoperative mastication pain and contour deformity. The scalp was closed in layered fashion over a short subgaleal drain.

Emergence from anesthesia was uneventful, with symmetric motor examination and fluent speech. The patient was transferred intubated to the neurosurgical high-dependency unit for standardized cerebrovascular surveillance, with systolic arterial pressure maintained between 110 and 140 mmHg (median 124 mmHg) and mean arterial pressure between 75 and 90 mmHg (median 82 mmHg) under euvolemic fluid management, while end-tidal CO_2_ was held within 35–40 mmHg (median 37 mmHg) to preserve a relaxed brain and stable intracranial hemodynamics. Pupils were equal and briskly reactive upon arrival. As anesthesia was lightened, purposeful antigravity movement was present in all four limbs with intact cough and gag reflexes and no neglect or aphasia, and extubation proceeded without incident following a successful spontaneous breathing trial with stable gas exchange and preserved airway protection. Analgesia was delivered in a non-sedating multimodal regimen to maintain the sensitivity of serial neurological examinations, and nausea as well as cough were proactively suppressed to avoid hypertensive surges. Intermittent pneumatic compression was instituted in the operating room and chemoprophylaxis for venous thromboembolism commenced on postoperative day one after confirmation of stable hemostasis and a dry wound. Serum sodium remained within 138–141 mmol/L, hematocrit within 39–42%, and capillary glucose within 90–130 mg/dL. No leukocytosis or inflammatory marker rise was observed. The subgaleal drain produced 55 mL in the first twelve hours and 15 mL in the subsequent twelve hours, was removed on postoperative day one without reaccumulation, and the incision remained flat, non-erythematous, and dry throughout.

Neurological examinations were performed hourly for the first twelve hours, every two hours for the next twelve hours, and every four hours thereafter, and no interval change was detected at any time point. Critically, the preoperative micro-asymmetries resolved in a manner that could be quantified at the bedside: by postoperative day one, rapid alternation between unrelated semantic categories proceeded without the previously documented micro-pause at category switches. Confrontation testing with randomized high-contrast micro-targets showed symmetrical reaction times, with the side-to-side difference in the right inferior quadrant measuring less than 50 ms and within examiner variance. Sustained isometric testing of right shoulder abductors and elbow flexors demonstrated no fatigability over timed holds. Rapid alternating forearm pronation–supination remained crisp and error-free beyond twenty cycles. Two-point discrimination at the index fingertips was symmetric at 3.0–3.2 mm, and sequential fine-touch localization over the right radial hand distribution no longer exhibited delayed spatial identification. Deep tendon reflexes were symmetric, and instrumented sampling of brachioradialis latency showed no side-to-side discrepancy beyond measurement noise. Gait, including extended tandem walking, was steady with midline foot placement without the preoperative seventh-step deviation. The postoperative course remained free of fever, wound complications, seizures, or electrolyte disturbances. Pulmonary hygiene, early mobilization exceeding fifty meters on postoperative day one, and progressive diet advancement proceeded uneventfully. A short prophylactic course of levetiracetam was completed without adverse effects, and there were no clinical seizures. Speech-language evaluation documented fluent spontaneous speech with intact naming and repetition and preserved praxis, and focused neuropsychological screening did not reveal deficits in processing speed or executive function.

Immediate postoperative neuroimaging was not obtained, consistent with the institutional protocol in neurologically stable patients after uncomplicated microsurgical aneurysm clipping. Predefined triggers for urgent imaging—any new focal deficit, sustained escalation of headache, decline in level of consciousness, or refractory hypertension—did not occur during inpatient monitoring. The patient was discharged on postoperative day three with instructions regarding blood pressure control, wound care, graded activity progression, and scheduled outpatient visits at two weeks, six weeks, and three months. At interval clinic assessments, he reported complete resolution of the preoperative vascular-pattern headaches, absence of exertional provocation, and restored confidence in balance and sustained fine-motor endurance during overhead tasks. The neurological examination remained normal and unchanged from the inpatient baseline. At three months, non-contrast cranial CT (Figure 3) demonstrated the expected clip artifacts at the MCA bifurcation and proximal M1 with preserved parenchymal attenuation throughout the superior division territory and within the internal capsule and dorsal basal ganglia, without encephalomalacia, hydrocephalus, or extra-axial collection, radiologically corroborating durable exclusion of both aneurysms with intact branch and perforator perfusion. Functional outcomes were congruent with the radiology and clinical course: the patient met criteria for complete independence with a modified Rankin Scale score of 0 and National Institutes of Health Stroke Scale (NIHSS) of 0 at discharge and again at the three-month follow-up.

The durable resolution of preoperative micro-asymmetries, together with structurally preserved dominant-hemisphere parenchyma on the three-month CT, supports effective clip reconstruction with maintained superior-division branch flow and uncompromised lenticulostriate perfusion.

Taken together, the convergent bedside findings, the precise preoperative vascular definition, and a cautious, anatomy-first microsurgical reconstruction allowed this patient to recover without new deficits, with resolution of the subtle preoperative asymmetries and structurally preserved dominant-hemisphere parenchyma at three months, a picture consistent with maintained superior-division branch flow and intact lenticulostriate perfusion. We present this case not as a prescription for others but as a transparent account of how a careful neurological examination shaped the imaging strategy, how the imaging constrained the operative plan, and how each intraoperative step was paced to protect veins, branches, and perforators. The favorable course may reflect the faithful application of established principles rather than any innovation—vein-respecting Sylvian fissure work, deliberate proximal control, and clip placement that accepted time and exposure in exchange for perforator safety. We recognize the limits of a single case, the absence of immediate postoperative vascular imaging, and the relatively short follow-up, and continued surveillance is planned. With the patient’s informed consent and within institutional protocols, we offer this experience in the hope that its small practical details may be useful when similar anatomy and clinical signals are encountered.

## 3. Discussion

This tandem configuration—an anterosuperiorly projecting, broad-neck middle cerebral artery bifurcation sac that partially incorporated both M2 origins lying only a few millimeters distal to a short dorsal M1 fusiform dilation entwined with lenticulostriate mouths—required a plan able to settle two distinct problems without taxing perforator reserve or compromising branch caliber. We elected to address the fork first in order to secure an immediate neck solution that left both M2 lumina round, pink, and freely mobile, then to remodel dorsal M1 longitudinally with every perforator ostium kept continuously in view. The subsequent abatement of the preoperative micro-asymmetries and the structurally preserved dominant-hemisphere parenchyma on interval imaging are consistent with that intent and suggest that branch geometry and perforator inflow were maintained where it mattered most.

For the bifurcation sac, endovascular options were examined and declined for anatomy-specific reasons rather than by reflex. Intrasaccular occlusion would have depended on perfect device sizing and seating across a saddle already sharing neck real estate with both M2s. Even a modest under-sizing or tilt in a tight fork risks residual neck, branch impingement, or the need for adjunctive constructs that add complexity without guaranteeing better ostial freedom. Stent-assisted coiling would have brought dual antiplatelet exposure into a language-dominant fork and placed a coil mass immediately adjacent to partially incorporated ostia, a combination that narrows the safety window when subtle caliber changes can carry disproportionate cortical cost [16]. Flow diversion at the fork has matured and can induce progressive occlusion, yet the price is borne by a jailed cortical branch that must accept a period of neointimal remodeling with a non-zero probability of stenosis or occlusion. The choice of which M2 to scaffold materially influences both the thrombosis vector and the branch that carries the physiologic burden [17]. In this setting, a direct neck reconstruction offered an immediate and observable answer about both M2 orifices without transferring risk to a delayed biological process.

For the proximal M1 fusiform segment, the constraint was perforator sovereignty. Covering clustered lenticulostriate origins with a new neointimal plane, straightening the parent trunk, or altering local shear can each exact neurologic currency in the posterior limb of the internal capsule and dorsal basal ganglia. Flow diversion across this short-segment dilation would have spanned precisely those perforator mouths that could instead be left unroofed. Parent-artery occlusion with bypass remained a prepared alternative, but the segment length and the tight grouping of ostia permitted a longitudinal clip parallel to flow to close exclusively on the diseased wall while leaving each perforator mouth visible, round, and free of torque. Managing the fork first and the trunk second also limited the chance that maneuvers at one lesion would destabilize the other in a corridor bounded by veins and eloquence. To place our patient-level decisions in context, we compiled key 2024–2025 studies directly relevant to MCA bifurcation and proximal M1 aneurysm management. Table 1 aims to concisely summarize cohorts, techniques, antiplatelet regimens, occlusion and safety outcomes, and branch–perforator behavior so readers can map these data to anatomies comparable to ours. It is offered as a focused reference point rather than a comprehensive review.

Because no intraluminal implant was placed, neither perioperative antiplatelet agents nor systemic anticoagulation were required, which helped keep hemorrhagic risk low in the dominant hemisphere while preserving the sensitivity of the neurological examination. Venous thromboembolism prophylaxis followed a simple sequence of immediate mechanical compression and low-dose chemoprophylaxis from the first postoperative day once hemostasis was secure. Had an endovascular route been selected, current practice would have favored individualized platelet inhibition with thoughtful attention to single-agent strategies where appropriate and guided dual therapy when hyporesponsiveness was demonstrated, ideally with point-of-care platelet function testing to avoid both under- and over-inhibition. At the MCA, where jailed-branch and perforator physiology compress the margin for error, that individualization is not a formality but part of the safety plan.

Revascularization remained available as insurance against either an uncooperative fork or a parent trunk unwilling to accept a remodeling clip. Low-flow superficial temporal artery to MCA bypass, including double-barrel constructs, can protect one or both divisions while neck work proceeds. Side-to-side M2 anastomosis, interposition grafting, or intracranial-to-intracranial reimplantation extend the palette when a branch origin must be re-sited away from a hazardous neck. In this patient, deliberate basal bone work, a vein-respecting Sylvian opening, and strict arachnoid-plane dissection created the necessary slack and line of sight to accomplish neck closure and parent-wall reshaping without adding bypass ischemia or suture-line risk.

Several failure modes that recur in MCA surgery were anticipated and addressed explicitly. Occult branch stenosis can arise when clip blades cross a shared neck with even slight obliquity, turning a round ostium into a subtle oval that the eye may tolerate but the physiology may not. Orienting a fenestrated clip perpendicular to the long axis of the neck, encircling the superior M2 through the window, and inspecting both ostia for color, contour, and mobility before and after closure mitigated that hazard. Perforator capture is the characteristic error in dorsal M1 disease when a fusiform segment is treated as if it were a saccular neck. Applying a short, gently curved blade longitudinally, advancing closure in small increments, and keeping every lenticulostriate mouth continuously visible, with residual bulges softened by tangential, low-amperage bipolar rather than deep coagulation, protected the ostia without transmitting thermal injury through the wall. Venous injury in the fissure can nullify arterial success by trading a clean reconstruction for a venous infarct. Keeping arachnoid sleeves intact on the superficial Sylvian system, redirecting the working corridor when veins constrained the trajectory, and avoiding torsion on bridging tributaries preserved venous outflow. Adherent hemostatic materials were kept off the reconstructed arterial segments to prevent delayed tethering and mural irritation that complicate re-entry and follow-up [28].

Ischemic management paired technical discipline with quiet physiology. Temporary occlusion at the fork was limited to a single interval of two minutes and thirty seconds, and the M1 reconstruction required a shorter interval comfortably under two minutes. Control was applied in a proximal-to-distal sequence and released in the reverse order so that reperfusion proceeded from the periphery inward. Mean arterial pressure was held in the high–normal range and end-tidal carbon dioxide in the mid-thirties to support collateral flow without vasodilatory edema. Brain relaxation relied on staged cisternal release and fissure splitting, and no fixed retractors were used so that the frontal lobe fell away under gravity and the temporal operculum rested on a soft interface rather than on hardware [29]. Blood loss was low, no transfusion was required, crystalloid administration remained euvolemic, and urine output reflected stable renal perfusion. These apparently mundane details help ensure that technically exact clip work is not undermined by preventable systemic swings [30].

Imaging fidelity directly influenced the operative plan. Biplane angiography, with sufficient temporal resolution to visualize early venous transit, steered clear of interpretive pitfalls attendant on anatomy of the neck being obscured by overlapping arteries, and thin-slice reconstructed rotational datasets elucidated the saddle geometry, angles of M2 take-off, and short perpendicular courses of the lenticulostriates across the fusiform segment. Reconstructions of bone and vascular structures were used to estimate the distance to the limen insulae, the spatial relation to the sphenoid ridge, and the line of sight that would keep the superficial Sylvian veins lax in the operating period. Through this series of minor predictions, there are less adjustments to be made under the microscope and less temptation to accept non-optimal clip vectors in the insulated space [31].

Recently, the precision frontier in MCA aneurysms is marked by technology dual insignificance in operative performance. The use of computational fluid dynamics, formerly alone a matter of academic modeling, has entered into the operatively exploitable sphere. Recent papers described patients who combined CFD patient modeling with deployed real-world bifurcation flow-dividers, vividly unveiling how the design of which M2 to support determines as much the thrombosis of the sac as the stress and strain imparted to the jailed branches [32,33]. That same year, the CFSS reached maturity and introduced an understandable, reproducible metric at the level of each branch as to the caliber and degree of change of diffusion-weighted lesions, which has occurred after cortical-branch coverage, having given birth to the only standard form of vocabulary that accounts for jailed-branch physiology [8]. Imaging parallel to that generated has made surveillance equally bricks and mortar: photon-counting CT angiography (PC-CTA), which has been judged successful in both translational and clinical studies, is able to resolve in-device lumina and branches adjacent to the clips applied, with a resolution that approaches that of the usual catheter angiography, while the radiopaque artifact is diminished [34,35]. These new aids make workmanship no more than a measurable vascular science, explicating linkages of clip geometry, wall shear redistribution, and flow adaptation, in one undivided, data-informed continuum—without to any extent displacing the anatomical judgment, which is still the final arbiter in dominant-hemisphere surgery [36].

Adjunct technologies were deliberately not used, and their absence is described as a deliberate choice rather than a virtue. Neuronavigation, micro-Doppler, fluorescence angiography, indocyanine-green videoangiography, and intraoperative angiography can all reduce uncertainty about inflow, outflow, and clip purchase and are integrated to good effect in hybrid environments. Vessel-wall imaging and photon-counting computed tomographic angiography now improve post-treatment visualization of in-device lumina and clip-adjacent branches. In this case, careful corridor preparation, vein-respecting dissection, and strict visual–tactile criteria for vessel health allowed progress without those tools, but their availability remains a useful redundancy, and they are reasonable complements wherever they are standard.

Emerging biomechanics and wall biology offer a coherent backdrop for the clinical pattern observed here. Patient-specific flow modeling at MCA forks emphasizes that wall shear stress, oscillatory indices, and wall-motion variance are spatially heterogeneous and evolve with subtle geometric changes, so that the placement of a device or the angle of a blade can redistribute stress fields between branches in ways that are not intuitively obvious [37]. Molecular profiling of aneurysm domes in recent series has outlined macrophage-centered inflammatory programs and matrix remodeling signatures that correlate with structural plasticity and may help explain why some regions remodel silently while others declare themselves through symptoms or growth [38]. Although these findings are not yet prescriptive at the bedside, they render plausible the tight mapping between the patient’s load-sensitive association-cortex latency and fine-motor fatigability and the anatomic sites where hemodynamic stress and wall remodeling were most likely to interact.

Follow-up policy reflected a preference for clinically driven surveillance when recovery is smooth. Immediate postoperative computed tomography was not obtained because the examination remained stable and our institutional criteria reserve urgent imaging for predefined triggers, such as a new focal deficit, sustained escalation of headache, impaired consciousness, or refractory hypertension. Outpatient surveillance included structured clinical review, non-contrast computed tomography at three months, and a plan for lumen-focused vascular imaging at twelve months by computed tomographic angiography or catheter angiography depending on access and image quality needs, an approach that balances radiation and contrast exposure against the likelihood that early imaging would alter management in an otherwise uneventful course.

Resource considerations were acknowledged without allowing them to outrank anatomy. Open reconstruction avoided device cost, prolonged antiplatelet therapy, and the cadence of angiographic surveillance that follows implants. Those signals are system-dependent and interact with local logistics and patient comorbidity, and they are best used to plan responsible pathways once the anatomical imperatives and the patient’s priorities have set the direction.

Limitations are clear. This is a single-patient experience from a microsurgical program, there was no immediate postoperative angiography, and the three-month follow-up relied on non-contrast computed tomography. Longer-horizon luminal confirmation is planned. The favorable course should not be generalized beyond anatomies that share the same fork geometry, perforator layout, and venous constraints, and the description is offered to make the reasoning and the micrometric steps transparent rather than to advocate a universal pathway.

The patient’s account is consistent with the objective narrative, with the pulsatile headaches and brief lapses in processing speed resolving after surgery and confidence returning for sustained overhead tasks, and with a steady return to work and family routines without new neurological limitations. In aggregate, the lessons are modest and practical rather than grand, namely, that in the MCA tree, small structures and short times hold most of the truth, that millimeters at a fork and microns at a perforator mouth carry more weight than any single modality label, and that when decisions and hands respect those scales, the nervous system often finds the room it needs to recover.

## 4. Conclusions

We conclude by reemphasizing the measure that presided in this operation over every detail—that dictated by anatomy. At the fork, a common neck united both origins of M2 but a moment away from a short dorsal dilation of M1, in which were lodged lenticulostriates. There were two lesions, one delicate, sufficiently ample in space to preserve all about which care was concerned. Aiming at the fork first and at the trunk second yet keeping each perforator mouth before the eyes continuous, rhythm of the circulation in the dominant hemisphere was restored to its normalcy without increased risk of ischemia and without dependence on auxiliary technology. What dictated the result was not innovation, but attention: a conscientiousness for methods and a respect and waiting tolerance for the exquisite architecture that today, as ever, determines how the brain is supported and recovered. The patient’s quiet, symmetrical recovery is less a triumph than a confirmation that small accuracies still matter.

What should travel forward is simple. Let subtle, load-sensitive signs count as data, not curiosities, and let them set the pace before infarction fixes the narrative. Describe outcomes in the language of branches and perforators, not only sacs and devices—roundness, mobility, and shape at an ostium are the currency that predicts how a life feels after surgery. Use imaging not as an accessory but to answer a few crucial questions about the neck saddle, take-off angle, and perforator course; then, make the corridor gentle enough that those answers remain true under the microscope.

For the near future, there is already a good effect. There is potential in individual flow motion simulations of patients, which are focused on two or three possibilities as to yarn’s vectorial principles before the first arachnoid is opened. In noninvasive observation of the lumen, it can be seen whether or not confusion of the side walls of a recreated wall or jail branch has begun, without a catheter. There is promise in recreated strategies of chosen platelets according to the physiology of branches and perforators, and not the labels attached to furniture of filling, or in tolerance maps of the lenticulostriate area, which substitute for the anecdotal evidence and are attained by something measurable. They do not demand that the exercise of our judgment should be sacrificed, but they demand that our judiciousness be shown in a better light. If we can emphasize one thing, it is restraint in the interest of precision.

When everything trends toward accuracy in a true sense of dimension—the millimeter scale at the common saddle, the micron in the mouth of a perforator, the moment’s time under temporary staphylococcal infection—the nervous system does much of the rest. We hope that the detail given above will make the next such case recognized a little easier to deal with, a little safer to go through, and, if good luck is on our side, just as unobserved in the patient.

## Figures and Tables

**Figure 1 diagnostics-15-02678-f001:**
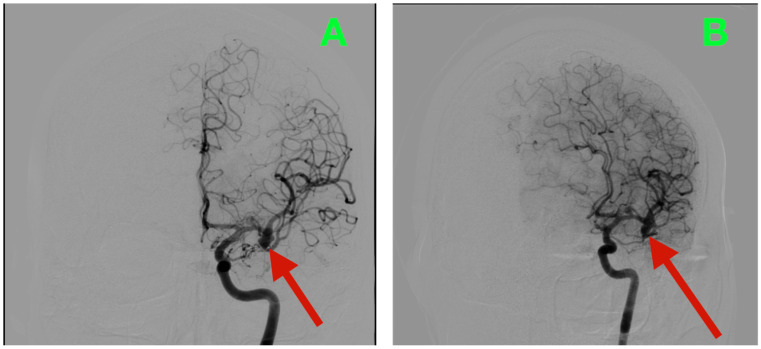
Preoperative DSA of the left internal carotid artery. (**A**) Anteroposterior view demonstrating a broad-necked saccular aneurysm arising from the M1 bifurcation, incorporating the origins of both superior and inferior M2 trunks, with an anterosuperior dome projection toward the opercular–insular interface. Proximal to the bifurcation, a fusiform dilation of the dorsal M1 segment is visible within the lenticulostriate perforator belt. (**B**) Lateral projection further delineating the dome axis in relation to the insular apex and overlying opercular cortex, as well as the proximity of the fusiform lesion to perforators supplying the posterior limb of the internal capsule and dorsal basal ganglia.

**Figure 2 diagnostics-15-02678-f002:**
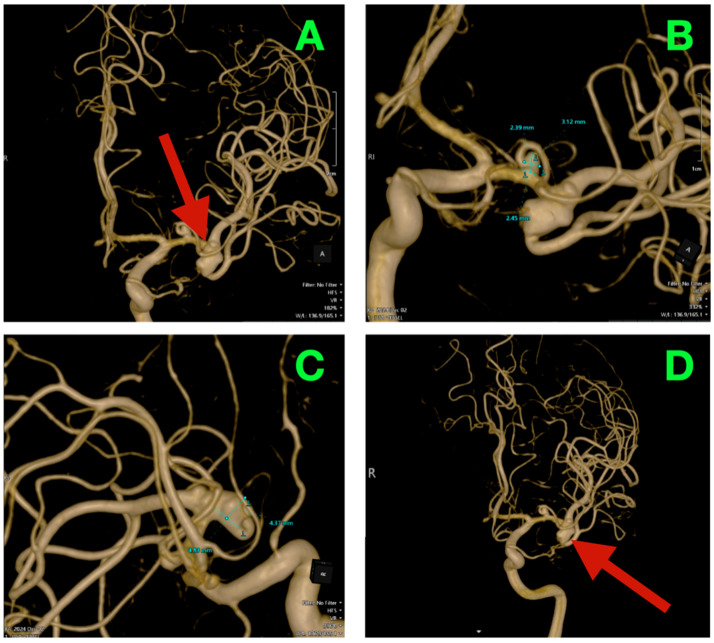
Three-dimensional rotational angiography and CTA volumetric reconstruction of the left MCA complex. (**A**) 3DRA, left internal carotid artery injection, anterolateral oblique projection. The bifurcation aneurysm arises at the M1 division point, with the broad neck partially incorporating both superior and inferior M2 origins. Note the anterosuperior dome projection toward the mid-insular cortex and the tight angular divergence of the M2 trunks (“tight fork” configuration), leaving minimal tolerance for clip blade malrotation. (**B**) 3DRA, orthogonal craniocaudal projection. The dome’s spatial relationship to the superficial Sylvian venous complex is evident, with the shortest opercular vein lying within 3 mm of the aneurysmal wall. The fusiform M1 dilation is visible proximally on the dorsal surface, directly within the lenticulostriate perforator belt. (**C**) CTA bone–vascular fusion, lateral projection. Depth mapping from the sphenoid ridge shows the bifurcation aneurysm positioned approximately 17 mm deep, corresponding to the mid-third Sylvian fissure corridor, while the fusiform aneurysm lies deeper (~23 mm), just proximal to the genu of the MCA. (**D**) CTA bone–vascular fusion, inferolateral projection. The lenticulostriate perforators arising from the fusiform M1 segment are visualized with submillimetric resolution, demonstrating perpendicular dorsal take-off toward the anterior perforated substance, underscoring the extreme vulnerability of these vessels to even minimal parent vessel displacement during bifurcation clip application.

**Figure 3 diagnostics-15-02678-f003:**
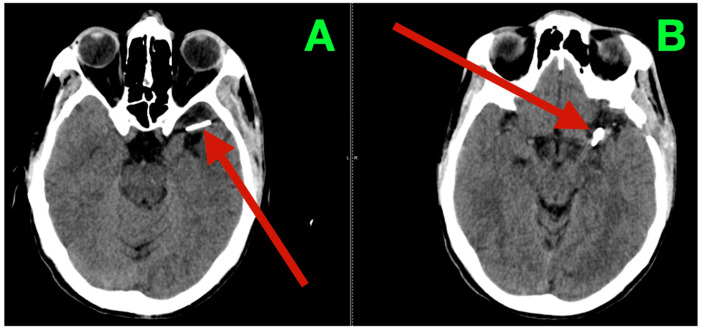
Three-month postoperative non-contrast cranial CT. (**A**) Axial non-contrast CT at the level of the left MCA bifurcation/insular apex, showing the expected clip artifact centered on the bifurcation–proximal M1 complex, with preserved cortical ribbon along the opercular–insular corridor, normal attenuation of the internal capsule and dorsal basal ganglia, and no peri-clip hypodensity, hemorrhage, or mass effect. (**B**) Axial non-contrast CT at a slightly higher level through the superior division territory, demonstrating intact suprasylvian cortex and corona radiata without encephalomalacia or gliotic change. Ventricles and basal cisterns are normal in size and configuration, midline is preserved, and there is no extra-axial collection.

**Table 1 diagnostics-15-02678-t001:** Summary of recent (2024–2025) studies relevant to MCA bifurcation and proximal M1 aneurysm care. For each study, we note the cohort, anatomy, technique or drug strategy, main occlusion and safety outcomes, branch/perforator effects, and follow-up. The selection is pragmatic rather than exhaustive and is intended to help relate our patient to current primary data. Results are shown as reported by the authors, and differences in imaging protocols, antiplatelet regimens, definitions, and follow-up intervals should be kept in mind.

Focus	Study/Year	Cohort and Anatomy	Approach	Main Outcomes	Branch/Perforator Insight	Follow-Up	Ref.
Clipping vs. WEB at MCA bifurcation	BMC Neurol (2025), IPTW + PSM	288 unruptured MCA WNBAs (WEB 37/Clip 251)	Standard clipping vs. WEB	Adequate occlusion 97.4% vs. 76.1%; similar morbidity	M2 incorporation → surgical advantage	12 months (angio + clinical)	[11]
Flow-diverter porosity and ischemia	Front Neurol (2024)	79 MCA bifurcation aneurysms	p48 MW HPC/p64 MW HPC/p64 classic	Occlusion 88.9–100% (device-dependent)	Porosity drove DWI lesions and branch caliber loss (~7.8% ischemic)	Mid-term	[6]
Which M2 to scaffold in FD cases	JNIS (2025), CFD + clinical	20 MCA bifurcation aneurysms	CFD-guided FD placement	Hemodynamic benefit depends on chosen M2	Quantified jailed-branch stress vs. sac thrombosis	Early	[18]
Grading covered branch behavior (CFSS)	JNIS (2025)	FD-treated MCA aneurysms	Caliber-Flow Status Scale + DWI	CFSS predicts flow/caliber change	First reproducible branch-level metric	Early–mid	[8]
FD reconstruction near LSA belt	J Brain Sci (2024)	Proximal M1 aneurysms	FD ± adjuncts	Favorable occlusion; ischemic events dominant	Highlights perforator risk in LSA-rich zones	Variable	[19]
M1-specific ischemic risk (multicenter)	Front Neurol (2025)	Mixed MCA/M1 aneurysms	FD families across centers	Acceptable occlusion; center variability	M1 confirmed as independent risk site	Multicenter	[20]
Predictive grading for post-clip ischemia	J Clin Neurosci (2024)	222 pts/251 MCA aneurysms	Microsurgical clipping	Risk score for post-clip ischemia	Branch incorporation and neck geometry = key predictors	≤9 years	[21]
Surgical trend analysis (2 decades)	Acta Neurochir (2024)	Elective anterior circulation	Technique evolution	↓ Ischemia and neurologic deficits over time	Reflects refined branch handling	Longitudinal	[22]
Photon-counting CTA (preclinical)	Diagnostics (2024)	Swine + phantom FD models	PC-CTA vs. conventional	Clearer stent + branch visualization/less artifact	Enables early noninvasive stenosis detection	Experimental	[23]
Photon-counting CT (clinical)	JNIS (2025)	Post-FD human datasets	Dedicated stent kernels	Improved lumen clarity vs. CT	Quantitative branch caliber monitoring	Clinical	[24]
Coated FD under single-agent prasugrel	AJNR (2024)	Unruptured aneurysms (p64 MW HPC)	SAPT vs. DAPT	Comparable efficacy; lower bleeding risk	SAPT may limit hemorrhage in select cases	Early–mid	[25]
Low-dose ticagrelor strategy	JNIS (2024)	SAC/FD aneurysms	Half-dose ticagrelor + ASA vs. clopidogrel	Less hemorrhage with maintained protection	Supports individualized antiplatelet therapy	Cohort	[26]
Neck-level flow disruptor vs. FD (in silico)	Biomech Model Mechano (2024)	2 MCA bifurcation models	Contour vs. FD simulation	Greater inflow + WSS reduction w/out branch coverage	Suggests lower jailed-branch penalty	Simulation	[27]

## Data Availability

The data presented in this study are available upon request from the corresponding author. The data are not publicly available due to patient privacy and ethical restrictions, as they originate from a real clinical case.

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
