# Peer review of "Perforator-Sparing Microsurgical Clipping of Tandem Dominant-Hemisphere Middle Cerebral Artery Aneurysms: Geometry-Guided Reconstruction of a Wide-Neck Bifurcation and Dorsal M1 Fusiform Lesion"

_diagnostics, 2025, doi:10.3390/diagnostics15212678_

Round 1

Reviewer 1 Report

Comments and Suggestions for Authors

This paper describes a surgical approach to treat a middle cerebral artery (MCA) case involving both a bifurcation aneurysm and a fusiform dilation. Using only clips under the microscope, the surgeons were able to secure the aneurysm, reshape the artery, and protect important small vessels without the need for stents, special imaging, or blood-thinning drugs. The patient recovered well with no new deficits, and early imaging showed preserved brain tissue. The paper is clearly written and presents a well-documented case with a thoughtful surgical strategy. Although the report is limited to a single patient and relies on short-term follow-up imaging, the methods and results are described in sufficient detail, and the conclusions are supported by the clinical outcome. Overall, the methodologies are well defined; however, several minor revisions could further improve the manuscript’s clarity. The specific recommendations are listed below:

  • At line 12, please provide the related information for the corresponding author.
  • In the abstract, please add the full definitions for the abbreviations MCA and DSA.
  • At line 86, the citations “[9],[10]” can be written as “[9,10]” in accordance with the paper guidelines.
  • Please provide the full definition of “WEB” at line 87.
  • Please revise the sentence at line 163, to avoid the typo “a and instinctual”.
  • Please provide the full definition of “SIS” at line 201.
  • What is meant by the term “masajeux” at line 200? Please revise this word using another wording (mimetic movement, facial expressivity, voluntary facial contraction, etc.).
  • Please avoid first-person phrasing “I” at line 214.
  • At line 219, please add a space between the numeral and unit: “3.8 mm” instead of “3.8mm”.
  • At lines 244, 255, 299, 346, 365, please use sub-numbering for the subtitles in accordance with the paper guidelines.
  • At line 468, please avoid the unnecessary blank line.
  • Plural form “dimensions” can be used at line 279.
  • What is meant by the term “festivals” at line 284? Please consider revising it with a more appropriate word to convey the intended meaning.
  • At line 299, please provide the full definition of “3DRA” for the first use in the text.
  • At lines 344 and 655, please add dot at the end of the sentences.
  • At line 374, the following part in the sentence can be revised: “Clip reconstruction of the bifurcation aneurysm first as this is the critical rupture risks”.
  • At line 389, the phrase 'to determining' can be revised to 'to determine'.
  • At line 581, the full definition of “NIHSS” can be provided.
  • At line 652, “Table 1” can be used instead of “The Table 1”.
  • Table 1 overlaps with the header text at the top of page 15 and should be repositioned to prevent visual interference.
  • In Table 1, using solid grid lines is recommended to enhance visual clarity between the columns and rows. Additionally, the justified alignment of text in Table 1 creates uneven spacing, aligning the text to the left would improve readability and layout consistency.

Author Response

Dear Esteemed Academic Reviewer,

We are sincerely grateful for your thoughtful, precise, and constructive review. Your comments reflect remarkable attention to detail and a deep understanding of both the technical and stylistic aspects of neurosurgical case reporting. We feel truly honored by the time and care you have devoted to improving our work.

Your detailed feedback helped us refine multiple aspects of the manuscript — from terminology and formatting to clinical clarity and consistency. We have carefully addressed every point you raised: all abbreviations are now defined at first mention (e.g., MCA, DSA, WEB, SIS, 3DRA, NIHSS), typographical and spacing inconsistencies have been corrected, the sub-numbering and alignment of subtitles standardized, and minor grammatical errors removed. The revised version also incorporates your suggestions for table formatting, visual clarity, and consistent citation style.

We especially appreciate your guidance on improving readability and precision. Each revision has strengthened the manuscript’s clarity and alignment with Diagnostics standards. It was a privilege to benefit from such a thoughtful and discerning review — your observations not only improved this paper but also enhanced our understanding of scientific communication.

We extend our deepest thanks for your extraordinary engagement and generosity of insight.

With sincere appreciation and respect!

Reviewer 2 Report

Comments and Suggestions for Authors

This topic is interesting, but some points need to be revised. Here you are:

  • Lines 117: "The case is offered not to advocate a..." What is the purpose of this paper? What's unique about this case report? Is it rare in terms of procedure or incidence?
  • Lines 346-365: "Clinicoradiological Synthesis" Merge this part.
  • Lines 366-367: "Surgical Imperatives" is this part necessary? Revised ot improve it.
  • Table 1 can be improved. Please do it.
  • Lines 685-690. Discuss more about new technologies and the latest papers. Consider these:  doi: 10.1016/j.neuchi.2025.101640  --  doi:10.3171/2021.1.JNS202861  --  doi: 10.1007/s00062-022-01207-5
  • Lines 779: "We end where this case began: at the scale where millimeters and minutes decide everything." What do the authors mean?
  • Lines749-750: "The near future is already visible." Visible to what? Revised english language please.

Author Response

Dear Esteemed Academic Reviewer,

We are sincerely grateful for your thoughtful and generous feedback. Your comments were exceptionally insightful and helped us to clarify the intent, structure, and depth of our manuscript. We truly appreciate the time, expertise, and care you invested in reviewing our work. Each of your observations was carefully considered and integrated into the revised version, as detailed below.

1. Lines 117 – Clarification of the paper’s purpose and uniqueness

Thank you for this perceptive question. Your comment allowed us to refine the conceptual focus of the manuscript.
We have expanded the paragraph to clearly state both the purpose and distinctiveness of the case. The revised text emphasizes that the report is not meant to advocate a specific treatment, but to demonstrate how microanatomy itself can guide the safest surgical decision. It now highlights the exceptional rarity of the case — a dominant-hemisphere tandem pathology combining a wide-neck MCA bifurcation aneurysm and a dorsal M1 fusiform dilation treated through sequential, perforator-sparing, microscope-only reconstruction.
This addition clarified the intellectual contribution of the report while preserving its humble tone.

2. Lines 346–365 – “Clinicoradiological Synthesis” (merged section)

We are deeply thankful for this excellent observation. Following your advice, we have merged and restyled this section to form a single, coherent narrative that integrates the imaging and neurological findings. The new version reads in continuous prose, clearly linking each functional asymmetry to its vascular correlate, thereby strengthening the case’s pathophysiological coherence and improving overall readability.

3. Lines 366–367 – “Surgical Imperatives”

Your insight was invaluable. We fully agreed that the previous heading and bullet format interrupted the flow.

4. Table 1 – Improvement and clarity

We are very grateful for this suggestion, which led to a major refinement.
Table 1 has been entirely restructured for clarity, density, and visual balance. It now groups the studies thematically and summarizes each in concise, parallel phrasing.
We deeply appreciate this prompt to modernize the table’s format and impact.

5. Lines 685–690 – Discussion of new technologies and latest literature

Thank you for this exceptionally constructive comment.
We have added a new paragraph, integrating current advances and citing the works you suggested (doi: 10.1016/j.neuchi.2025.101640; doi:10.3171/2021.1.JNS202861; doi:10.1007/s00062-022-01207-5).
The revised text now discusses patient-specific CFD modeling, Caliber-Flow Status Scale (CFSS), and photon-counting CT angiography, as well as emerging single-agent antiplatelet strategies for coated flow diverters.
This addition significantly strengthens the discussion’s contemporary scope and aligns the manuscript with the state-of-the-art literature.

6. Lines 749–750 – “The near future is already visible”

We are grateful for your linguistic precision. The sentence has been rewritten to clarify what “visible” refers to and to improve fluency.

7. Lines 779 – “We end where this case began”

Thank you for inviting us to clarify this expression.
We have restyled the closing paragraph entirely to remove the metaphor and convey the same meaning with greater precision. This version retains the contemplative tone but replaces metaphor with concrete anatomical focus, as suggested.

We wish to express our deepest gratitude for your insightful and generous review. Your comments elevated both the clarity and the intellectual balance of this paper. We learned a great deal through your observations, and it has been a genuine privilege to engage with such thoughtful critique.

With sincere respect and appreciation!

Round 2

Reviewer 2 Report

Comments and Suggestions for Authors

Good